# An Automatic Baseline Correction Method Based on the Penalized Least Squares Method

**DOI:** 10.3390/s20072015

**Published:** 2020-04-03

**Authors:** Feng Zhang, Xiaojun Tang, Angxin Tong, Bin Wang, Jingwei Wang

**Affiliations:** State Key Laboratory of Electrical Insulation & Power Equipment, Xi’an Jiaotong University, Xi’an 710049, China; zfczl1314@stu.xjtu.edu.cn (F.Z.); tongangxin@stu.xjtu.edu.cn (A.T.); wangbin168@stu.xjtu.edu.cn (B.W.); wangjingwei@stu.xjtu.edu.cn (J.W.)

**Keywords:** automated baseline correction, infrared spectra, penalized least squares

## Abstract

Baseline drift spectra are used for quantitative and qualitative analysis, which can easily lead to inaccurate or even wrong results. Although there are several baseline correction methods based on penalized least squares, they all have one or more parameters that must be optimized by users. For this purpose, an automatic baseline correction method based on penalized least squares is proposed in this paper. The algorithm first linearly expands the ends of the spectrum signal, and a Gaussian peak is added to the expanded range. Then, the whole spectrum is corrected by the adaptive smoothness parameter penalized least squares (asPLS) method, that is, by turning the smoothing parameter λ of asPLS to obtain a different root-mean-square error (RMSE) in the extended range, the optimal λ is selected with minimal RMSE. Finally, the baseline of the original signal is well estimated by asPLS with the optimal λ. The paper concludes with the experimental results on the simulated spectra and measured infrared spectra, demonstrating that the proposed method can automatically deal with different types of baseline drift.

## 1. Introduction

Spectroscopic analysis technology such as Raman and infrared spectroscopy has been widely used in many fields because of its high sensitivity, rapid analysis speed, and nondestructive features [1,2,3,4,5]. However, any spectra measured by a spectrometer usually have a wandering baseline due to environmental factors, such as light source, temperature, humidity, and vibration during long-time continuous work [6]. When the baseline drift spectrum is used for quantitative and qualitative analysis, the prediction accuracy of the model is reduced. Hence, it is necessary to correct the baseline of spectra before subsequent analysis [7,8].

Recently, researchers have paid much attention to baseline correction and proposed several methods [9,10,11,12,13,14,15,16,17,18,19,20,21,22,23,24,25]. These baseline correction methods are mainly iterative polynomial fitting [9,10,11], derivative method [12,13], wavelet transform method [14,15,16], iterative averaging [17,18,19], and penalized least squares [20,21,22,23,24,25]. Iterative polynomial fitting is performed by an iterative strategy where, in each iteration, a polynomial is used to obtain the fitted baseline. The minimum value between baseline and signal is taken as the new signal, and when the fitted baseline no longer changes or the changes are minimal, a satisfactory baseline is obtained. However, an iterative polynomial tends to obtain a boosted baseline in the peak regions and cannot be used in situations where the background changes drastically. The derivative method amplifies the noise signal during baseline correction. When the spectrum has high frequency noise, the corrected signal is distorted. Therefore, the spectrum should be smoothed before applying the derivative method. When the wavelet transform method is used to correct the baseline of the spectra, the spectra signals are decomposed into low and high frequency signals. Therefore, the baseline signal with low frequency is removed by the wavelet transform and reconstruction. However, it is difficult to select the optimum wavelet basis, decomposition level, and threshold of the wavelet coefficients. Iterative averaging tends to overestimate the baseline in peak regions when the width of the spectral peak area is large. In addition, it is not suitable to fit the baseline when the spectra are overlapped.

Baseline correction methods based on penalized least squares are widely used in various types of spectral preprocessing because of its fast speed and its ability to avoid peak detection. The main idea is to balance the fidelity and smoothness of the fitted baseline. The application of penalized least squares in the field of baseline correction was first proposed by Eilers, who designed the asymmetric least squares (AsLS) method [20] to correct the baseline of various spectra in 2003. For the AsLS method, the weight vector updates iteratively, and if a signal is above the fitted baseline, then a small weight is assigned to achieve automatic interpolation fitting. On the other hand, a large weight is given when a signal is below a fitted baseline. Subsequently, researchers have developed several improved algorithms based on the AsLS algorithm. Zhang et al. considered that the AsLS algorithm has two turning parameters, λ and p. Moreover, the weight can only be set to p and 1-p, which results in the same weight of the peak regions as that of the noise regions, so the estimated baseline obtained by this method is not optimal. In this respect, an adaptive iteratively reweighted penalized least squares (airPLS) method [21] was proposed that further improves the performance of the baseline correction. The core idea of airPLS is to assign different weights according to the difference between the signal and the fitted baseline; in addition, only one parameter needs to be optimized. He and Zhang proposed an improved asymmetric least squares (IAsLS) [22] baseline correction method. Compared with the AsLS and airPLS methods, the performance of the fitted baseline is further improved. However, when a spectrum is corrupted with additive noise, both the airPLS and IAsLS methods tend to obtain an underestimated baseline. In order to perform baseline correction in noisy signals, an asymmetrically reweighted penalized least squares smoothing (arPLS) [23] method was proposed by Park, which uses the iterative method to adaptively obtain weights according to the generalized logic function. Although the arPLS can obtain a satisfactory baseline in the no-peak regions, a boosted baseline is obtained in the peak regions. In order to obtain a satisfactory baseline in the whole spectrum region, we previously proposed an approach called the adaptive smoothness parameter penalized least squares (asPLS) [24]. However, the above baseline correction methods have one or more tuning parameters, which limits the application of real-time online data analysis.

According to our experiments, the smoothing parameter λ has a great influence on the performance of baseline correction, but in all of above methods λ is set randomly by the users. Therefore, a baseline correction method of extended range based on penalized least squares (erPLS) is proposed in this paper by selecting the optimal parameter λ of asPLS. The proposed method is compared with the improved modified multi-polynomial fitting (I-ModPoly), morphological weighted penalized least squares (MPLS) [25], and iterative averaging (IA) methods in the simulated spectrum and the real infrared spectrum, and the comparison results confirm that the proposed method can better handle diverse baseline types automatically.

## 2. Materials and Methods

### 2.1. Simulated Data Generation

For the presentation and validation of the proposed method, several mathematical spectra were generated to simulate real infrared spectra in different practical applications. The simulated spectra data consisted of an analytical signal and different types of baseline and random noise, which is described as the following equation:y(v) = s(v) + b(v) + n(v)(1)
where y(v) is the simulated data, s(v) denotes the analytical spectral data, b(v) represents the baseline, and n(v) displays the random noise. The spectra scanned by FTIR are actually a convolution of the real spectra and the apodization (window) function, so the shape of the spectra is similar to that of the Gaussian function. Therefore, the analytical signal is generated by several Gaussian peaks and mathematically presented as follows:(2)$$s(v)=2e^{-{\left(\frac{v-100}{20}\right)}^{2}}+e^{-{\left(\frac{v-200}{20}\right)}^{2}}+2e^{-{\left(\frac{v-400}{40}\right)}^{2}}+e^{-{\left(\frac{v-500}{30}\right)}^{2}}+4e^{-{\left(\frac{v-800}{50}\right)}^{2}}+0.5e^{-{\left(\frac{v-1000}{15}\right)}^{2}}+e^{-{\left(\frac{v-1100}{20}\right)}^{2}}+1.5e^{-{\left(\frac{v-1200}{20}\right)}^{2}}$$
where Equation (2) is the generated analytical signal, and the numbers in the equation and even the number of Gaussian peaks can be modified according to the application in different situations. The spectra baseline is prone to linear or curved drift. Thus, we simulated linear and curved baselines to evaluate the performance of the proposed baseline correction method: (1) a first-order linear baseline and (2) a sine baseline. The two types of baselines are given by the following formula:(3)$$\{\begin{array}{l} b_{1}=-0.01+2\times 10^{-3}x\\ b_{2}=\sin(\pi x/1200) \end{array}$$

Random noise was generated using MATLAB’s awgn function. In order to verify the performance of the proposed method in different noises, two different levels of noise were generated—the signal-to-noise ratio (SNR) of the high noise spectrum was set to 25 dB and that of the low noise spectrum was set to 30 dB. The simulated spectra were obtained by adding the baseline signal, the noise signal, and the analytical signal, which are shown in Figure 1. To distinguish the simulated spectra easily, the high noise spectra was moved up by two.

### 2.2. Infrared Spectra Acquisition

The mid-infrared spectra were scanned using a Spectrum Two Fourier transform (FTIR) spectrometer, produced by PerkinElmer (Waltham, MA, USA). The optical path was 10 cm. The spectrum resolution was set to 1 cm^−1^ and 8 scans were used for each spectrum. The spectral range and apodization function were set to 400~4000 cm^−1^ and Norton–Beer medium, respectively. Thus, there were 3601 variables for each spectrum. Methane and n-butane were selected as target analytes, and the absorption spectra are shown in Figure 2.

It is obvious from Figure 2 that the baseline drifted more seriously as concentration decreased. This is because the absorption signal became smaller when the concentration decreased. Thus, baseline drift is prone to occur with a slight change of environmental factors.

### 2.3. The Proposed Method: erPLS

The methods based on penalized least squares obtain a satisfactory baseline by turning the parameters. For the AsLS method, baseline drift is eliminated by adjusting the asymmetry p and smoothness λ. For the airPLS, arPLS, and asPLS methods, a satisfactory baseline is obtained by changing the smoothness parameter λ. However, these methods must be optimized by one or two parameters; more importantly, these parameters have a great impact on the results, which limits the application of these methods in the field of real-time online spectra analysis. In order to overcome the shortcomings of the above methods, a baseline correction method called extended range based on penalized least squares (erPLS) is proposed. The core idea of erPLS is to select the optimal parameter λ of the asPLS method. Before introducing the proposed method, three parameters are defined as follows: selected wavenumber range Ω, and added Gaussian peak width W and height H. We selected Ω as the wavenumber range at the end of the spectral signal used for linear fitting and we used W and H to generate a Gaussian peak. In general, the length of Ω is recommended to be set to one-twentieth of the spectral length N, whereas W is usually assigned one-fifth of N, and H is set equal to the maximum of the ordinate values of the spectra signal. A detailed description of the erPLS algorithm is as follows:

Step 1: Linear fit. A first-order polynomial is used to perform a linear fit on the spectral data in the selected wavenumber range to obtain a linear regression coefficient. The fitting equation is given as follows:(4)$${\hat{y}}_{i}=ay_{i}+b\;(i\in \Omega )$$
where a and b are first-order regression coefficients and ${\hat{y}}_{i}$ is the fitted value of the spectral signal in the selected wavenumber. 

Step 2: Linear expansion. Extended signal **y**_e_ is obtained by linear expansion of the end of the spectra signal according to the regression coefficient, and the length of **y**_e_ is W.

Step 3: Signal addition. Add a Gaussian peak signal **y**_g_ to the extended signal, the width of Gaussian is equal to W/2, and the height is H. The new extended signal is marked **y**_eg_, **y**_eg_ = **y**_e_ +**y**_g_. In this way, the new spectral signal **y**_new_ is composed of two parts—one is the original spectra signal **y** and the other is the extended signal **y**_eg_, **y**_new_ = [**y**, **y**_eg_]. The progress of steps 1–3 is shown in Figure 3.

Step 4: Calculate the RMSE values for the extended range. The extended range RMSE is denoted as RMSE_e_. The whole spectrum **y**_new_ is corrected by the asPLS method. RMSE_e_ is calculated by the following equation:(5)$$RMSE_{e}=\sqrt{\frac{\sum_{i=1}^{W}{\left[\hat{y}(i)-{\hat{b}}_{e}(i)\right]}^{2}}{W}}$$
where ${\hat{b}}_{e}$ is the fitted baseline of the extended range obtained by the asPLS method, ${\hat{y}}_{i}$ is defined in step 1, and *W* is the length of the extended range. For this case, different RMSE_e_ values are obtained by turning λ in the asPLS method, and are shown in Figure 4. From the figure, one can see that the optimal λ is 10^9.6^ because this value obtained the lowest RMSE_e_.

Step 5: Select the optimal λ. The λ with the lowest RMSE_e_ is selected as the optimal smoothing parameter.

Step 6: Output the final fitted baseline of the original signal y. The baseline of the original signal **y** is corrected by the asPLS method with optimal λ. 

### 2.4. Parameter Setting

In this research, all experiments were performed on MATLAB (R2016b, The Math Works, Natick, MA, USA). In order to evaluate the performance of the above baseline correction methods, the root-mean-square error (RMSE) was adopted: (6)$$RMSE=\sqrt{\frac{\sum_{i=1}^{N}{\left[b(i)-\hat{b}(i)\right]}^{2}}{N}}$$
where *b* is the simulated baseline that is denoted as the real baseline, $\hat{b}$ is the estimated baseline, and *N* is the length of the original spectra signal. As mentioned above, the critical step of the proposed erPLS method is to find the optimal λ for the asPLS method. In order to observe how the smoothness parameter λ affects the performance of the arPLS and asPLS methods, λ was turned from 10^3^ to 10^12^ with varying in a log scale. The experiments were tested in a simulated spectrum with a sine baseline and with low noise. The RMSE values obtained by two methods are shown in Figure 5.

From Figure 5, one can see that the parameter λ had a great impact on the arPLS and asPLS methods. For the arPLS method, RMSE gets a small value when 10^6.6^ ≤ λ ≤ 10^8.3^, while for the arPLS method, RMSE keeps low when 10^8.4^ ≤ λ ≤ 10^9.9^. For convenience, this paper refers to 10^8.4^ ≤ λ ≤ 10^9.9^ as the proper range. A careful look at Figure 5 reveals that the RMSE value is greater than 1 when λ is less than 10^4^, and less than 0.05 when λ is in the proper range. Therefore, we developed a method to select parameter λ of the asPLS method.

In the erPLS algorithm, three parameters—the selected wavenumber range Ω and the added Gaussian peak width W and height H—are important and should first be optimized. Experiments of sine baseline simulated spectra with low noise were conducted to observe the influence of the above parameters.

In order to research how the Gaussian peak width W and height H affect the performance of the erPLS method, the erPLS was calculated using different values of W and H made with a fixed Ω of the wavenumber range 1151~1200 cm^−1^. The parameter W was changed from 200 to 400 with an interval of 10, while the parameter H was turned from 3 to 6 with an interval of 0.2. The different λ were computed with the various pairs of W and H, and the three-dimensional figure of λ vs. W and H is given in Figure 6.

It is obvious from Figure 4 that λ is in the proper range (10^8.4^ ≤ λ ≤ 10^9.9^) when W is varied from 200 to 400 and H from 3 to 6, which confirms that the erPLS is robust to the parameters W and H.

To investigate the relationship between the parameter Ω and the erPLS algorithm’s performance, the following settings were used: Ω = 1141~1200 cm^−1^, Ω = 1151~1200 cm^−1^, and Ω = 1161~1200 cm^−1^, where W and H are set to 200 and 5, respectively. The relation between Ω and erPLS algorithm’s performance is shown in Figure 7. It is clear from Figure 7 that the fitted baselines almost overlap, which also shows that the erPLS is robust to the selected wavenumber range Ω. Through these experiments, we can reach the conclusion that the above three parameters can be set automatically according to the spectrum signal when applying the erPLS method.

## 3. Results

### 3.1. Experimental Simulated Data

To evaluate the performance of the erPLS method, some lately proposed baseline correction methods, namely I-ModPoly, MPLS, and IA, were used to simulate the infrared spectra for a comparison with erPLS. All of the above four methods have one or more parameters that must be set before performing the baseline correction. The parameters of erPLS are automatically set according to the spectrum signal. For the I-ModPoly method, the polynomial order was turned from 1 to 10 with increments of 1. The optimal order was determined after a few loops. For the MPLS, the critical smoothness parameter λ was turned from 10^3^ to 10^9^ with varying in a log scale, the order of difference penalties was set to 2, and the half window w size of the morphological structuring elements was set from 10 to 200 with increments of 10. The optimal parameters λ and w were carried out by finding the minimum value of the RMSE in the MPLS method. For the IA method, we chose the optimum parameter threshold T according to reference [19]. The RMSE values obtained by the four baseline correction methods under optimal parameters are listed in Table 1.

From Table 1, among the four baseline correction methods, the erPLS method obtained the smallest RMSE values, the MPLS and I-ModPoly methods obtained relatively better RMSE values, while IA obtained the worst RMSE values. Hence, we can conclude that the erPLS method can obtain an improved result for linear or curved baselines compared to the I-ModPoly, MPLS, or IA methods, especially when the spectra are corrupted with high noise. It can also be seen from Table 1 that the RMSE values obtained in high noise were larger than those obtained in low noise. Therefore, we can confirm that noise has a great influence on the above baseline correction methods.

Detailed results of the baseline corrected by the four methods are shown in Figure 8. It is obvious from Figure 8 that the estimated baseline obtained by the erPLS method is more accurate than others. On the other hand, the I-ModPoly and MPLS methods are relatively worse, while IA obtained the worst result. The estimated baseline obtained by the erPLS almost overlapped the real baseline in both the linear and curved baselines with different noise levels. The I-ModPoly method obtained a boosted baseline, and the reason is that the calculated approximation of the noise level was too high. The fitted baselines handled by MPLS almost overlapped with the real baseline when the noise was low, while in high noise, the fitted baseline was lower than the real baseline. The IA method tended to pass through the lowest point of the spectrum, resulting in different degrees of underfitting in the whole spectrum. Therefore, it is necessary to perform a smoothing operation before applying the MPLS and IA methods. From the above experiments, we can conclude that the proposed baseline correction method shows better performance compared with the other methods.

### 3.2. Experimental Infrared Spectra

To verify the capability of the proposed erPLS method, six different concentrations of methane and n-butane spectra were used for the experiment. In order to clearly distinguish the performance of the above methods, the baseline-corrected spectra of n-butane with a concentration of 50 ppm and 100 ppm obtained by the four methods are shown in Figure 9.

According to Lambert’s law, the ideal absorption is zero in non-absorption bands. Therefore, for the infrared spectrum of n-butane, the performance of these methods can be evaluated by the degree of deviation from zero in the non-absorption region. As seen in Figure 9a, the MPLS and IA methods obtained a boosted corrected spectrum in the middle spectrum region (1600–2200 cm^−1^), while the I-ModPoly method overestimated the baseline in the left half of the non-absorption region (3000–3500 cm^−1^), resulting in a declining corrected spectrum in the left spectrum region. From Figure 9b, it can be seen that the MPLS and IA methods obtained a boosted corrected spectrum in the right spectrum region (600–1000 cm^−1^), while the I-ModPoly method obtained an underestimated baseline-corrected spectrum in the left half of the non-absorption region (3000–3500 cm^−1^); this phenomenon is more obvious when the concentration of n-butane is 50 ppm. The corrected spectrum performed by the erPLS method almost overlapped zero in the non-absorption region, whether the concentration of n-butane was 50 ppm or 100 ppm, which proves that the baselines corrected by the erPLS methods are more precise than those of the others. 

Six different concentrations of methane and n-butane spectra using the erPLS method were performed to show the capability of the proposed method, and the corrected spectra are displayed in Figure 10. Compared with Figure 2, it is evident that the absorption baseline-corrected spectra are close to zero in the non-absorption region, and look quite similar in the absorption regions, which further proves that the proposed method can successfully eliminate baseline drift.

Finally, it is worth mentioning that the optimal parameters cannot be selected by carrying out the minimal RMSE value, because we do not know the real baseline. Fortunately, we know the absorption in the non-absorption region. Therefore, for the I-ModPoly, MPLS, and IA methods, we chose the appropriate parameters according to many experiments. For the proposed erPLS method, the parameters Ω, W, and H were proven to have little effect on the results, which can be set automatically according to the spectrum signal. As shown in Figure 10, the drifting spectra were successfully corrected by the erPLS method; thus, we can conclude that the erPLS can automatically handle various types of baseline drift. 

## 4. Conclusions

In this research, an automated baseline correction method based on penalized least squares was proposed and the performance was evaluated by simulation and infrared spectra. The core idea of erPLS was to add a Gaussian peak in the extended range combined with the asPLS method to perform baseline correction automatically. 

For the simulated spectra analysis, the RMSE of the erPLS was smallest among the four baseline correction methods, and the estimated baseline almost overlapped the real baseline, whereas the other methods tended to deviate from the real baseline. The experimental results of the two different concentrations of n-butane infrared spectra proved that the erPLS method is more precise than the I-ModPoly, MPLS, and IA methods. Moreover, the six different concentrations of methane and n-butane spectra performed using the erPLS method also confirmed that the proposed method is more effective in dealing with different types of baselines. More importantly, the method can automatically correct the baseline without users having to set the parameters. These consistent results show that the erPLS method can correct various types of baselines automatically and can improve the accuracy of baseline estimation. Therefore, we hope that erPLS can be applied to unsupervised online analysis of FTIR systems in the future.

## Figures and Tables

**Figure 1 sensors-20-02015-f001:**
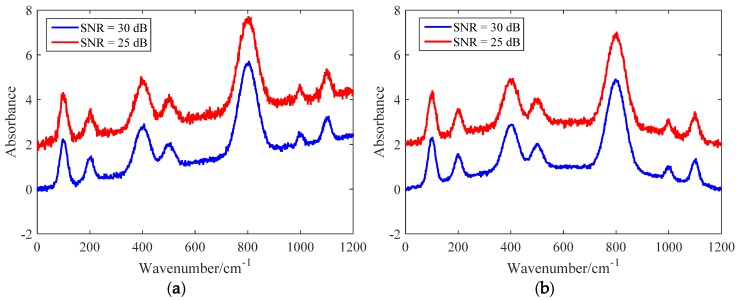
Simulated spectra with linear and sine baseline types in high noise and low noise: (**a**) linear baseline; (**b**) sine baseline. To distinguish the simulated spectra easily, the high noise spectra was moved up by two.

**Figure 2 sensors-20-02015-f002:**
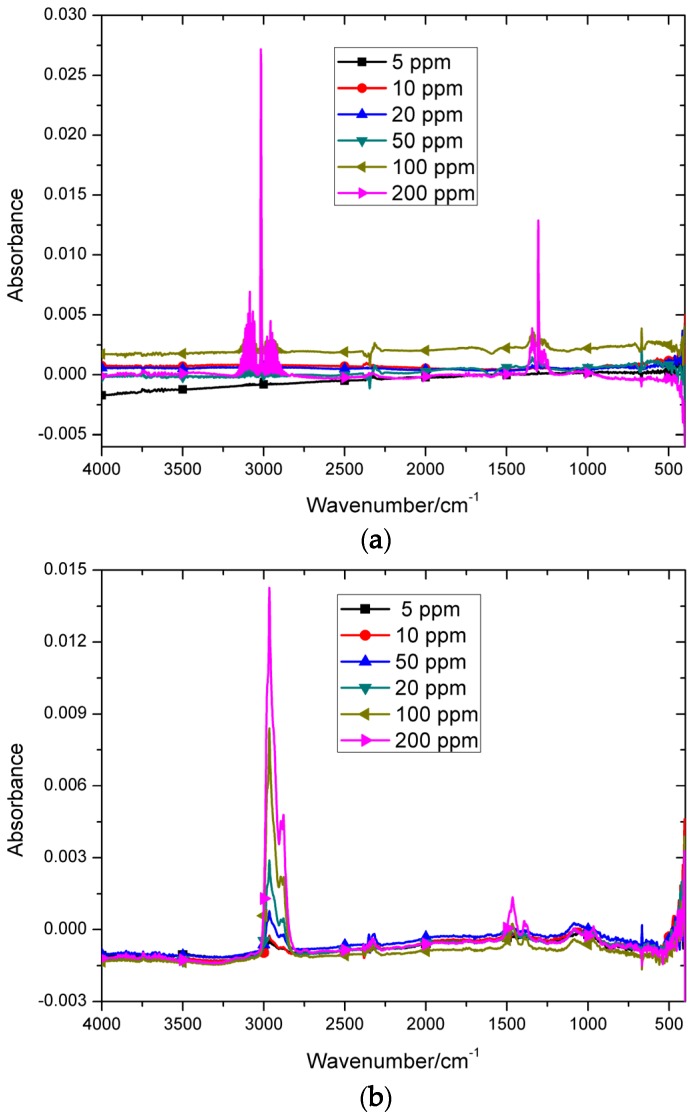
Six different concentrations of methane and n-butane spectra scanned by Fourier transform infrared spectrometer (FTIR): (**a**) methane; (**b**) n-butane.

**Figure 3 sensors-20-02015-f003:**
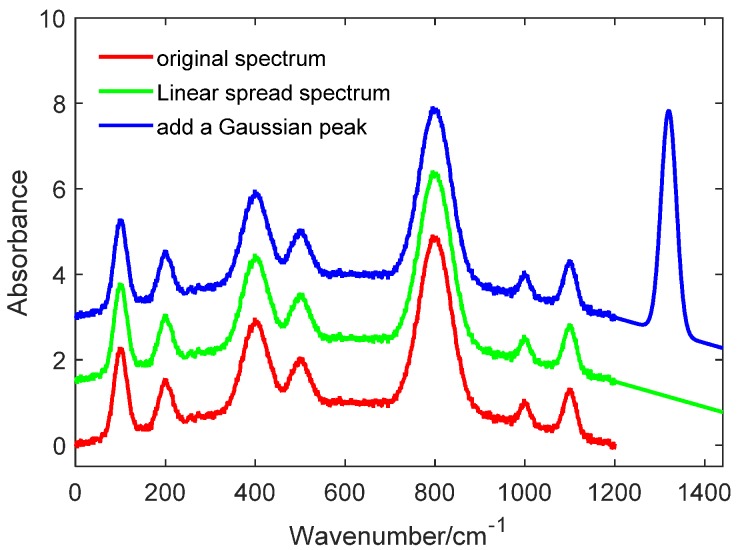
The original spectrum with linear extension and an added Gaussian peak.

**Figure 4 sensors-20-02015-f004:**
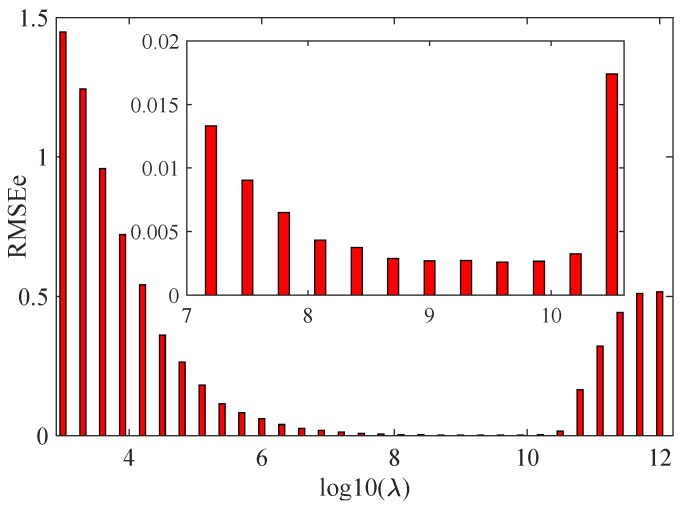
The relation between extended range root-mean-square error (RMSE_e_) and λ.

**Figure 5 sensors-20-02015-f005:**
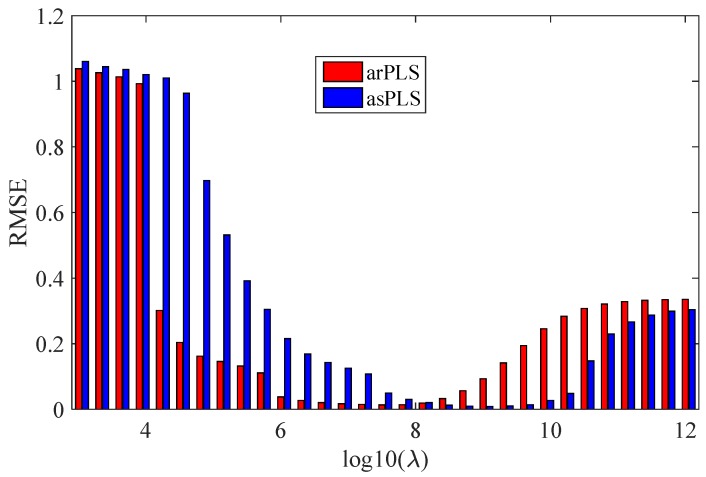
The RMSE of the asymmetrically reweighted penalized least squares smoothing (arPLS) and the adaptive smoothness parameter penalized least squares (asPLS) methods for different λ.

**Figure 6 sensors-20-02015-f006:**
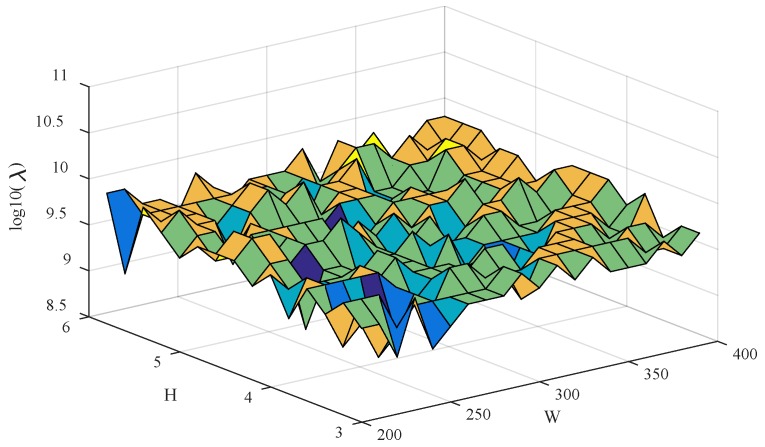
The selection λ between parameters W and H in the extended range based on penalized least squares (erPLS) algorithm.

**Figure 7 sensors-20-02015-f007:**
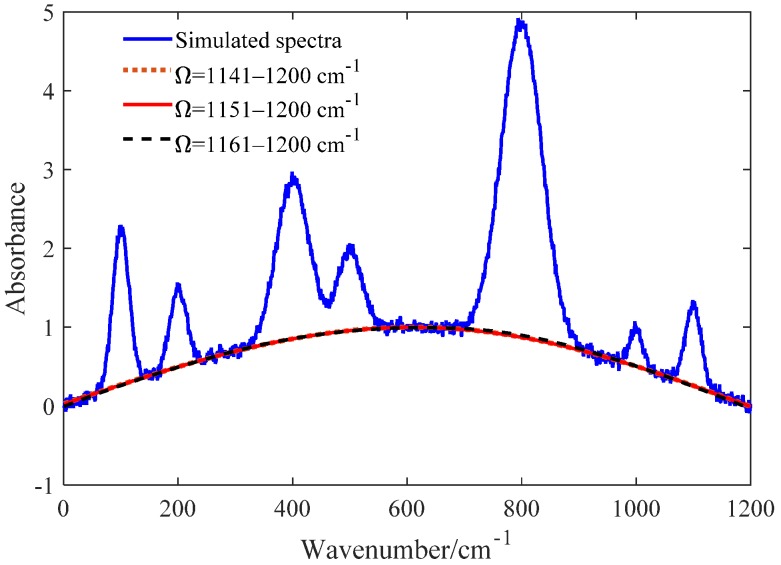
Comparison of the fitted baseline using the erPLS method between different selected ranges.

**Figure 8 sensors-20-02015-f008:**
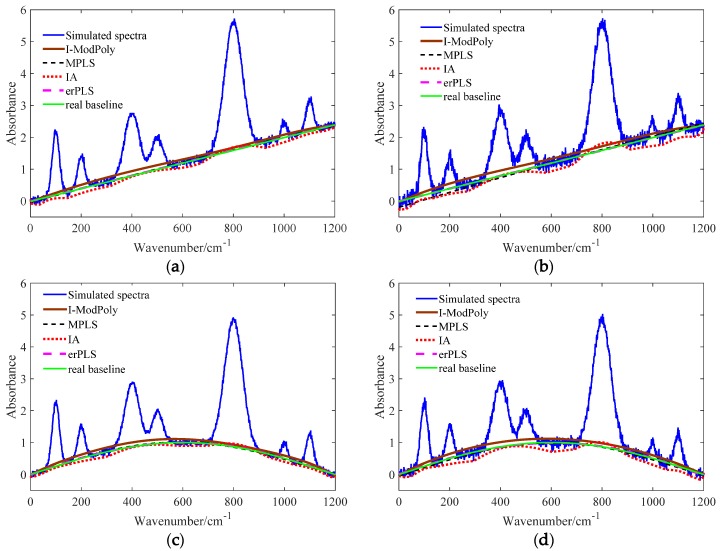
The fitted baselines with four methods in both high and low noise. (**a**) Linear baseline in low noise; (**b**) linear baseline in high noise; (**c**) sine baseline in low noise; (**d**) sine baseline in high noise. It can be seen that the erPLS method is more accurate compared with the I-ModPoly, MPLS, and IA methods.

**Figure 9 sensors-20-02015-f009:**
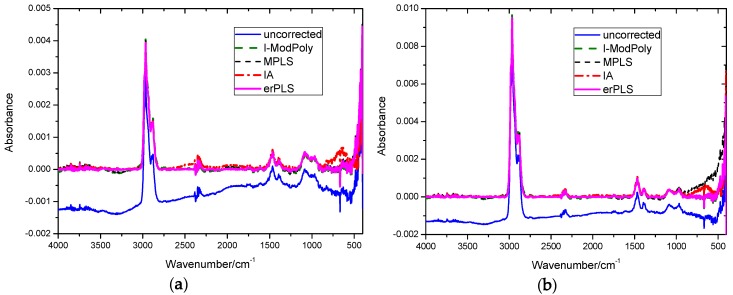
Baseline-corrected n-butane spectra by four methods: (**a**) the concentration of n-butane is 50 ppm; (**b**) the concentration of n-butane is 100 ppm.

**Figure 10 sensors-20-02015-f010:**
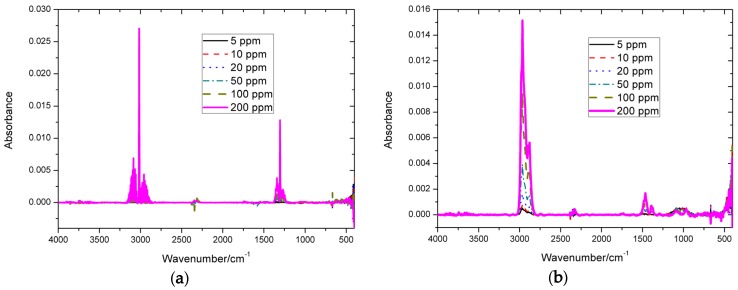
Baseline-corrected methane and n-butane spectra by the erPLS method: (**a**) the corrected spectra of methane; (**b**) the corrected spectra of n-butane. It is shown that the erPLS method successfully eliminates baseline drift.

**Table 1 sensors-20-02015-t001:** The RMSE values obtained by the four baseline correction methods.

Methods	Baseline Type (Low Noise)				Baseline Type (High Noise)			
	Linear		Sine		Linear		Sine	
	Optimal Parameters	RMSE	Optimal Parameters	RMSE	Optimal Parameters	RMSE	Optimal Parameters	RMSE
I-ModPoly ^1^	order ^5^ = 6	0.1043	order = 6	0.0790	order = 5	0.1315	order = 6	0.1164
MPLS ^2^	λ ^6^ = 10^8^ w ^7^ = 70	0.0131	λ = 10^7.6^ w = 80	0.0148	λ = 10^8.8^ w = 60	0.0764	λ = 10^7.6^ w = 70	0.0312
IA ^3^	T ^8^ = 10^−2.2^	0.1128	T = 10^−2.2^	0.0831	T = 10^−2^	0.1907	T = 10^−2^	0.1641
erPLS ^4^	-	0.0061	-	0.0036	-	0.0098	-	0.0109

^1.^ I-ModPoly—improved modified multi-polynomial fitting method; ^2.^ MPLS—morphological weighted penalized least squares method; ^3.^ IA—iterative averaging method; ^4.^ erPLS—the proposed method; ^5.^ order—polynomial order of the I-ModPoly method; ^6.^ λ—smoothness parameter of the MPLS method; ^7.^ w—half window parameter of the MPLS method; ^8.^ T—threshold parameter of the IA method.

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
