# Peer review of "An Automatic Baseline Correction Method Based on the Penalized Least Squares Method"

_sensors, 2020, doi:10.3390/s20072015_

Round 1

Reviewer 1 Report

The authors propose a new method of automatic baseline subtraction, based on penalized least square method improved with the addition of an artificial spectral range with one gaussian peak. The method is shown to give somewhat better results than another penalized least square method. Surprisingly, the quality of the baseline correction does not depend on the parameters of the added region.

The authors in some contexts (and in some not) compare this method with another improved penalized least square method they developed (asPLS). The comparison should be either full (including performance in figures 8 and 9) or not at all. This is funny since you claim that arPLS is an improvement of as PLS, but asPLS seem to perform better than ar-PLS according to the other paper of the authors https://doi.org/10.1080/00387010.2020.1730908. Can you comment on this? can you also comment on the fact that the "new" method you present in this paper is already mentioned in an already published paper of yours?

The references ion the introduction in many cases refer to papers dealing specifically with Raman spectroscopy. It is thus awkward to present this baseline subtraction method or other similar methods mentioned in the introduction as for FTIR specifically. Use the term "vibrational spectroscopy" where possible.

What das Spectrometer with a capital S on line 26 mean?

As references 1-4 I would expect reviews or textbooks, not random spectroscopy papers, as general facts are mentioned.

The penalized least square method (line 32/33) deserves a reference. The introduction below line 32 is generally low in references.

Please specify clearly that eq. 2 is the specific simulated spectrum used for testing and not a general formula for a simulated spectrum as follows from the text now.

Please justify the choice of the two baseline types you have tested (linear, sine).

Reviewer 2 Report

The article describes a novel method to make a baseline and I only have a small recommendation.

The authors do not indicate in what programming language they did this or if they used software, it would be good if the algorithms are published so that readers can use them more easily.

Reviewer 3 Report

The paper shows a follow up study on a new iterative approach for baseline correction. Although, this study appears to be interesting for a range of readers, there are some major issues that must be addressed before considering the paper any further:

1) the use of the English language should be substantially improved and it is recomended to the Authors to profread the paper by a native speaker. Without this issue carefully addressed, it is difficult to the Referee to understand the presented methodology in a great detail.

2) the spectral range in Fig 1 is not practical. It should be constrained to the one commonly used in FTIR, i.e. 4000-900 cm-1.

3) the Authors should show the efficacy of the Algorithm against various possible baselines faced in FTIR spectroscopy. It is advised to check how the algorith works for Mie-like baselines for various values of either refractive index and radious. With the current presentation, one example baseline, that is not so very common in FTIR (bio)spectroscopy is shown, though. Addressing this issue can extend potential readership. 

Round 2

Reviewer 1 Report

The authors have explained and addressed the comments and suggestions of the reviewer to an acceptable extent. The main issue was a misunderstanding of the ar-PLS vs. er-PLS methods on the side of the reviewer.

Reviewer 3 Report

Thank You for addressing my comments. The Referee can now recommend the paper formpublication.